# Pharmakokinetics of Mistletoe Lectins after Intravenous Application of a Mistletoe Product in Healthy Subjects

**DOI:** 10.3390/ph17030278

**Published:** 2024-02-22

**Authors:** Ann-Kathrin Lederer, Sabine Rieger, Michael Schink, Roman Huber

**Affiliations:** 1Center for Complementary Medicine, Department of Internal Medicine II, Medical Center-University of Freiburg, Faculty of Medicine, University of Freiburg, 79106 Freiburg im Breisgau, Germany; 2Department of General, Visceral and Transplant Surgery, University Medical Center, Johannes Gutenberg University, 55131 Mainz, Germany; 3Helixor Heilmittel GmbH, Fischermühle 1, 72348 Rosenfeld, Germany

**Keywords:** mistletoe lectins, phase I clinical trial, viscum album, pharmacokinetics, reversible liver injury

## Abstract

Mistletoe lectins (ML) have cytotoxic and immunomodulating properties, and subcutaneously applied mistletoe products (MP) containing ML have approval for supportive cancer treatment. MP are also given off-label intravenously, but data about pharmacokinetics are widely lacking. Therefore, the aim of our phase I trial was to evaluate the pharmacokinetics and safety of intravenously applied natural ML. Initially, 12 healthy male volunteers were planned to receive a single infusion of 2000 mg Helixor^®^ P. We had to terminate the study prematurely after the inclusion of eight subjects due to elevation of all subjects’ liver enzymes. ML was detected in all subjects after infusion. The mean half-life of serum ML was 7.02 ± 2.01 h. Mean alanine transaminase increased from 23 ± 6 to a maximum of 445 ± 260 U/L, and mean aspartate aminotransferase increased from 24 ± 3 to a maximum of 318 ± 33 U/L 72 h after infusion. Severity grading for drug-induced liver injury was mild. Participants did not suffer from any liver-specific symptoms and recovered completely. As a conclusion, the dose of 2000 mg Helixor^®^ P caused transient liver injury in healthy subjects and should, therefore, not be used for initial patient treatment. Liver enzymes should be monitored in patients receiving intravenous treatment with Helixor^®^ P.

## 1. Introduction

The white mistletoe (*Viscum album* L.) is a hemiparasitic plant that grows on deciduous trees (including poplars, willows, apple trees, and oaks) and conifers (firs and pines) [1]. Hemiparasites are capable of photosynthesis but obtain water, minerals, and organic nutrients from their host plant. Mistletoe products (MP) have presumably been used for centuries to treat several diseases such as cancer [1,2]. Commonly, watery extracts of the fresh plants are used as herbal remedies [3]. The efficacy of mistletoe has been the subject of intense debate for decades [4]. Recent studies indicate that MP improve quality of life and may improve outcome of cancer patients [5,6,7,8]. They are approved for subcutaneous use in Germany, Switzerland, Austria, Korea, and many other countries. MP are also given off label intravenously [9], but systematic evaluations of this treatment are widely lacking so far. In a previous dose finding study, we found excellent tolerability of up to 2000 mg of the MP Helixor^®^ P (P = from pine trees) applied once weekly intravenously [10]. Potential side effects of the highest dose were headache, flu-like symptoms, and fever. Two patients had mild transient alanine transaminase (ALT) elevation, and one patient had an allergic reaction [10]. Typical side effects after subcutaneous application of mistletoe also include local reactions and slight changes of inflammatory parameters in addition to the symptoms already described [3]. Steele et al. evaluated side effects after subcutaneous mistletoe application in almost 2000 patients and reported side effects in 15% of patients, with more than 90% classified as mild to moderate [11]. As described above, intravenous administration may cause fever and flu-like symptoms [9,10,12]. An observational study with Helixor^®^ A and M in Korea found good tolerability in 108 cancer patients in doses between 100 and 900 mg i.v. [13]. In another phase I study with 21 cancer patients, Helixor^®^ M was applied three times weekly; MTD was 600 mg i.v. (1800 mg/week). One grade 3 ALT increase and one grade 3 pain and dyspnea occurred [14]. A major compound of MP are mistletoe lectins (ML), which are cytotoxic, ribosome-inactivating proteins class 2 occurring in the three subtypes: ML-I, -II, and -III [3]. The molecular weight of ML-I–III is about 63 kDa. They have very similar biological properties and are composed of an N-glycosidase (A-chain) and a galactoside- or N-acetyl-galactosamine-recognizing lectin (B-chain) connected by a disulfide bridge [15,16,17]. The A-chain inhibits protein synthesis [18,19]. The B-chain binds to carbohydrate residues on the cell surface, thus entering the cell by receptor-mediated endocytosis and inducing apoptosis of the cell [19,20]. A recombinant type II ribosome-inactivating protein (rML) analogous to natural ML (nML) I revealed a short half-life of about 13 min in cancer patients [21]. Pharmacokinetics of nML so far have only been investigated after subcutaneous injection [22]. Median time to peak plasma concentration (T_max_) was two hours after subcutaneous injection. At the final examination (day 14 ± 3 after injection), serum nML concentrations were still detectable in 60% of the volunteers, with large interindividual variation. Detectability of nML in serum is, therefore, considerably longer than that of rML.

In summary, there are little data available to date on the intravenous application of MP, especially of MP containing nML. Therefore, with this clinical I trial, we aimed to investigate the pharmacokinetics of nML given intravenously in a commercial MP. We hypothesized that our methods and design allowed us to detect nML and to calculate half-life from our data.

## 2. Results

Overall, eight subjects were treated per protocol and analyzed. The characteristics of the subjects are shown in Table 1.

Mistletoe lectin was detected in all eight subjects after infusion of the IP. The plasma concentration–time curve is shown in Figure 1. As expected, elimination of ML followed an open two compartment model with a central (elimination 0–0.25 h) and a peripheral compartment (0.5–24 h). Levels below the detection threshold were found in one subject after 12 h, in six subjects after 18 h, and in seven subjects after 24 h. The mean half-life of serum ML after IP infusion was 7.02 ± 2.01 h. All pharmacokinetic parameters are shown in Table 2.

Initially, twelve male subjects were included into the study but only eight (two groups of four) were treated because we had to terminate the study prematurely due to elevation of liver enzymes, which occurred in all eight participants. Mean ALT increased from 23 ± 6 to a maximum of 445 ± 260 U/L 72 h after infusion of the IP; the absolute maximum was 892 U/L Mean AST increased from 24 ± 3 to a maximum of 318 ± 33 U/L 72 h after infusion of the IP; the absolute maximum was 474 U/L. LDH also substantially increased from 178 ± 18 to a maximum of 334 ± 84 after 72 h. GGT and AP remained within normal limits in all subjects at all time points. Liver function tests (bilirubin, prothrombin time, albumin) also did not deviate from the normal range in any of the subjects during the study. Severity grading for drug-induced liver injury according to the National Institutes of Health (NIH) was mild, as bilirubin remained normal [23]. Clinically, the increase in liver enzymes remained mostly inapparent, and classification according to the Common Terminology Criteria for Adverse Events (CTCAE) version 5.0 criteria for hepatobiliary disorders was, therefore, grade 1 (“asymptomatic or mild symptoms; clinical or diagnostic observations only; intervention not indicated”) [24]. Three subjects reported fatigue and flu-like symptoms (malaise and “feeling like the beginning of a flu”) only in the first days after infusion of the IP, which was possibly related to the study medication. None of the subjects developed fever. One subject suffered from headache. As liver enzymes had not returned to normal in six subjects at follow-up 14 ± 3 days after infusion, we asked these six subjects to check liver enzymes again 4–6 weeks later out of the study protocol. In all six subjects, ALT, AST, and LDH then returned to normal.

Overall, 78 unexpected events (UE) were reported, 40 of which had a potential causal relationship with the IP, mostly changes to blood values. In addition to the above-mentioned increase in liver enzymes, CRP and counts of neutrophil granulocytes slightly increased and lymphocyte count slightly decreased within 48 h after infusion, which can be interpreted as an acute phase reaction. Values returned to baseline levels at follow-up. Table 3 shows the course of relevant laboratory parameters.

Sixty-one UE recovered without consequences within the observation period, whereas seventeen UE (four with and thirteen without a causal relationship to the IP) were still ongoing at this time. None of the UE were classified as severe UE. Likewise, there was no severe unexpected adverse events (SUSAR). Besides the above-mentioned flu-like symptoms, in the group of UE with at least a possible causal relationship to Helixor^®^ P, redness of the infusion arm and venous irritation (phlebitis) occurred in one subject each.

## 3. Discussion

Here, we report the first study investigating pharmacokinetics of natural ML after the infusion of a commercial mistletoe preparation. Despite the fact that we had to terminate the study prematurely due to unexpected elevations of liver enzymes, we were able to calculate the pharmacokinetics parameter from eight subjects. Based on previous phase I studies, the sample-size was planned to be small (n = 12) [21,22]. Due to the premature end of the study, fewer subjects were included than initially planned. However, the pharmacokinetic results of the included subjects are very similar, so we assume that we can provide valid data on pharmacokinetics, despite the smaller-than-expected sample size.

Our study revealed that the half-life of natural ML (around 7 h) was longer than that of recombinant ML (13 min), which was previously reported by Schöffski et al. [21]. As recombinant ML and natural ML only differ in their tertiary structure, this considerable difference is most likely related to the tertiary structure of natural ML. After subcutaneous injection of a MP, ML could be detected in 60% of subjects’ sera until follow-up at day 14 [22]. Half-life was unable to be determined in this study. Here, it can be assumed that a subcutaneous deposit of ML caused a slow release.

In a previous phase I study in cancer patients, 2000 mg Helixor P^®^ was well tolerated [10]. One of nine patients treated with 2000 mg had a mild-to-moderate elevation of liver enzymes. At lower concentrations, none of the 21 patients included in the study developed increased liver enzymes. The strong increase in liver enzymes in all subjects of our study was, therefore, unexpected. In contrast to the previous study, we investigated young, healthy, physically active men. Their liver and immune system seemed to react much more sensitively to ML than that of older cancer patients. Liver cell damage due to high doses of mistletoe preparations has been discussed previously from single cases, but it has never been systematically proven in clinical studies [25,26,27]. The underlying mechanisms of liver cell damage after mistletoe application are unknown so far [28]. One publication discusses the possible effect of MP on cellular protein biosynthesis, but the results are difficult to compare as the MP used has a different ML and phytoconstituent content than the MP used in this trial [29]. Interestingly, other phytoconstituent of Helixor P^®^ have previously been described as potentially liver-protective, but again there is a lack of valid evidence [30,31]. Future studies should clarify what effects ML or other constituents could trigger in human liver cells.

As all subjects in our trial had no or only mild flu-like symptoms for a few days (no hospitalization, no treatment necessary); we did not report these adverse events as SUSAR. All symptoms completely disappeared, and the elevated liver enzymes normalized 6–8 weeks after infusion. Nevertheless, the dose used in this study appears to be not suitable for initial patient treatment due to its potentially hepatotoxic effect. Clinically, mostly lower dosages than 2000 mg Helixor^®^ P have been applied in cancer patients [9,32]. As liver enzymes have not been investigated in these studies and the database is currently too small to allow firm conclusions on which intravenously applied doses can cause elevation of liver enzymes, we suggest that liver enzymes should be monitored in patients receiving intravenous treatment with Helixor^®^ P.

## 4. Materials and Methods

### 4.1. Study Design

The present study was conducted between June and July 2022 and extends our previous dose finding study [10]. The study was a monocentric, open, non-controlled pharmacokinetic trial performed at the Center for Complementary Medicine at Freiburg University Medical Center in Germany. It was designed to investigate the pharmacokinetics of nML in the MP Helixor^®^ P within 72 h after a single infusion of 2000 mg. Safety and tolerability were monitored until follow-up 14 ± 3 days after infusion.

### 4.2. Recruitment

We recruited participants by placing advertisements on multiple notice boards at local universities, in student residences, and in public places in Freiburg, Germany. In addition, we posted advertisements on various social media platforms. Potentially eligible individuals were first screened by phone and subsequently invited for an in-person screening at our department for checking inclusion and exclusion criteria.

### 4.3. Inclusion and Exclusion Criteria

Only healthy non-smoking individuals aged 18 to 45 years and with a body mass index (BMI) between 18.5 and 28 kg/m^2^ were included. Blood pressure, pulse rate, body temperature, electrocardiogram (ECG), and routine laboratory parameters had to be normal. Criteria of exclusion included any sign of significant disease, regular intake of medication, clinically relevant allergies, positive HIV status, hepatitis B or C serology, alcohol or drug abuse, positive alcohol blood or drug urine test, blood donation in the last 3 months, difficult vein access, previous intake of mistletoe products in any form, and parallel participation in another study. Eligible participants had to be able to speak and understand German.

### 4.4. Investigational Product (IP) and Rationale

Helixor^®^ (Helixor Heilmittel GmbH, Rosenfeld, Germany) is a standardized, watery fresh plant extract from *Viscum album* L. that has been applied for subcutaneous use in Germany since 1982 as an anthroposophic remedy for cancer therapy. Helixor^®^ P is an extract from fresh mistletoe growing on pine trees. One ampoule of 2 mL contains the extract from 100 mg mistletoe, water for injection, and sodium chloride (99.91:0.09). Further compounds in the investigational product (IP) are oligo- and polysaccharides (predominantly ester derivates of D-galacturonane); phenylpropanglycosides (caffeic acid, ferulic acid, and sinapic acid); and flavonoids such as sakuranetin, rhamnazin, and isorhamnetin. Helixor^®^ P was selected because it has the highest ML content of all Helixor^®^ preparations. The dose of 2000 mg was selected because it was the highest tolerable dose in a previous phase I study [10]. Within the framework of toxicological testing, the cytotoxic effect of Helixor^®^ P was investigated on human hepatocytes by the MTT cytotoxicity test. Medium without addition of Helixor^®^ P served as a negative control, whereas 0.25 mg/mL sodium fluoride served as a positive control. Cytotoxic effects were demonstrated at concentrations of 0.05 mg/mL. The EC50 value of Helixor^®^ P was 0.015 mg/mL. In order to detect ML in the nanogram range, we opted for the highest possible dose. The expected ML concentration in our study was calculated from 1.248 µg ML/mL, corresponding to a total amount of 51.4 µg ML in 40 mL. Assuming a distribution in 3 L serum of a healthy male subject, this corresponds to 17 ng/mL. The detection threshold for quantitative ML determination of the selected ELISA was validated to ≥2.5 ng/mL.

### 4.5. Course of the Study

Individuals who successfully passed the screening examinations were hospitalized at 7 a.m. in the morning in groups of four. If no disease had occurred in the meanwhile, blood for baseline safety parameters and baseline ML concentration was taken. Thereafter, the investigational product was infused at a dose of 2000 mg (20 ampoules) in 250 mL of saline over exactly 150 min through a vein catheter, and 0, 0.25, 0.5, 0.75, 1.0, 1.5, 2, 3, 4, 6, 8, 10, 12, 18, 24, 36, 48, and 72 h after the end of the infusion, serum for the determination of ML concentration was obtained. Subjects were hospitalized until 24 h after the infusion; examinations 36, 48, and 72 h as well as 14 ± 3 days after infusion were ambulatory. Safety laboratory parameters were checked at 24, 36, and 48 h as well as 14 ± 3 days after infusion (Table 4).

### 4.6. Determination of Mistletoe Lectin in Sera

Concentrations of ML in sera of the subjects were determined with an enzyme-linked immunosorbent assay (ELISA) validated by Chimera Biotech GmbH, Dortmund, Germany, according to GLP standards. Monoclonal detection antibodies (coating: 5H8, detection: biotinylated anti-ML-A-5F5 b25) are specific for the A-chain of ML; they were provided by sifin diagnostics gmbh, Berlin, Germany. In the assay, only the total ML (ML I, II, and III) can be measured. As Helixor^®^ P contains only ML III, the obtained ML concentrations are equivalent to the ML III concentration. The linear measuring range determined in the validation process was 2.5–320 ng ML/mL serum.

### 4.7. Safety Parameters

Blood pressure, pulse rate, body temperature, ECG, changes in physical examination, and laboratory tests were the safety parameters. Coagulation: partial thromboplastin time (PTT), prothrombin time (PT); differential blood count; liver: ALT, aspartate aminotransferase (AST), alkaline phosphatase (AP), bilirubin, gamma-glutamyltransferase (GGT); kidney: creatinine, urea; pancreas: amylase, lipase; electrolytes: sodium, potassium, calcium; protein: albumin, total protein; lipids: cholesterol, triglycerides; C-reactive protein (CRP); creatin kinase; glucose; lactate dehydrogenase (LDH); uric acid.

### 4.8. Biosampling Details

Blood was taken by venipuncture and immediately centrifuged for 10 min at 2500× *g*. At least 2 mL of serum of each sample was aliquoted and kept frozen at −80 °C in the study center until the study was finished, and all samples were shipped on dry ice to Chimera Biotech in one batch. Safety laboratory analyses were performed by the accredited Central Laboratory of the University Medical Center of Freiburg (Institute of Clinical Chemistry and Laboratory Medicine, Medical Center—University of Freiburg, Freiburg im Breisgau, Germany).

### 4.9. Quality Assurance

This clinical trial was planned, conducted, and analyzed according to good clinical practice (GCP) regulations. Compliance to the study protocol, accordance between the original files and data in the case report form (CRF), and procedures in case of adverse events (AE) were, among others, checked during on-site monitoring. Data management was worked out by a data management group and described in a data management plan.

### 4.10. Ethical Approval

The ethical committee of the University Medical Center of Freiburg approved this trial (number 21-1693). The study was prospectively registered at the German Clinical Trial register (DRKS-S00027485). We obtained written and oral informed consent from all study participants.

### 4.11. Planning of the Sample Size

Analogous to previous pharmacokinetic studies with mistletoe lectins (Huber et al. (2010) = 15 subjects; Schöffski et al. (2005) = 10 subjects), we regarded 12 assessable subjects as adequate for this study [21,22].

### 4.12. Statistical Analysis

The study protocol defined three populations:-Subjects with a complete set of data until 72 h after infusion were the analysis population.-All subjects, irrespective of who received the infusion of the IP, were the safety population.-Subjects with major protocol violations were the drop-out population.

Pharmacokinetic properties of ML from Helixor^®^ P were characterized by the parameter elimination rate constant of the total phase 0–24 h (λ [/h]), half-life of the total phase 0–24 h (t1/2 [h]), speed constant for elimination from the central into the peripheral compartment (k0 [/h]), maximum serum concentration (C_max_ [ng/mL]), and area under the curve (AUC 0–24 h) [ng xh/mL]. They were calculated separately for each subject from the respective plasma concentration–time curve. The exponential equation for the total plasma curve speed constant C(t) is C(t) = 2.302^−0.0576t^ + 0.932^−0.0714t^.

Accordingly, for each time point t, the speed constant can be calculated.

## 5. Conclusions

Pharmacokinetics of natural mistletoe appear to be considerably longer than that of recombinant mistletoe. The dose of 2000 mg Helixor P^®^ causes transient liver injury in healthy subjects and should, therefore, not be used for initial patient treatment. Liver enzymes should be monitored in patients receiving intravenous treatment with Helixor P^®^.

## Figures and Tables

**Figure 1 pharmaceuticals-17-00278-f001:**
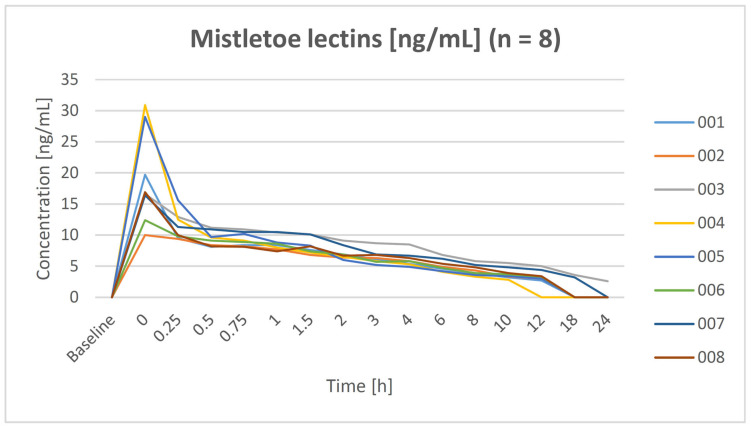
Course of serum mistletoe lectin concentration of each subject during the hospitalization (0–24 h).

**Table 1 pharmaceuticals-17-00278-t001:** Biometric data of the included subjects (n = 8).

Parameter	Mean	SD	Minimum	Maximum
Age (years)	29.60	5.68	21.00	36.00
Height (cm)	183.50	2.62	178.00	187.00
Weight (kg)	80.70	4.60	74.50	88.00
BMI (kg/m^2^)	24.00	1.54	21.80	26.30

**Table 2 pharmaceuticals-17-00278-t002:** Pharmacokinetic parameters.

Parameter	Mean	SD	Minimum	Maximum
λ [/h] (= elimination rate constant_(0–24 h)_)	0.11	0.03	0.07	0.17
t_1/2_ [h] (=half-life_(0–24 h)_)	7.02	2.01	4.18	10.38
ke [/h] (=speed constant for elimination from central into peripheral compartment	0.07	0.03	0.02	0.01
C_max_ [ng/mL] (=maximum serum concentration)	18.99	7.40	10.00	30.90
Area under the curve_(0–24 h)_ [ng × h/mL]	68.35	50.12	33.11	176.12

**Table 3 pharmaceuticals-17-00278-t003:** Mean ± standard deviation of selected laboratory parameters after infusion of 2000 mg Helixor P^®^ (n = 8).

Parameter	Screening	24 h	36 h	48 h	72 h	Follow-Up
Albumin (g/dL)	4.89 ± 0.21	4.69 ± 0.17	4.76 ± 0.18	4.53 ± 0.21	4.74 ± 0.15	n. m.
ALT (U/L)	23.38 ± 5.58	23.37 ± 5.58	236.40 ± 152.30	445.10 ± 260.20	84.25 ± 36.94	30.00 ± 6.16
AP (U/L)	77.38 ± 21.37	76.50 ± 18.98	76.75 ± 18.89	76.75 ± 17.30	72.00 ± 17.43	n. m.
AST (U/L)	23.88 ± 3.18	24.25 ± 2.12	223.30 ± 110.10	317.80 ± 133.30	34.63 ± 8.467	29.33 ± 5.61
Bilirubin (mg/dL)	0.57 ± 0.36	0.63 ± 0.27	0.49 ± 0.24	0.40 ± 0.22	0.59 ± 0.26	n. m.
CRP (mg/dL)	3.25 ± 0.71	3.70 ± 0.84	7.73 ± 3.18	6.16 ± 2.24	3.00 ± 0.00	n. m.
GGT (U/L)	17.00 ± 2.14	16.63 ± 2.67	18.38 ± 3.46	22.38 ± 5.61	27.00 ± 6.87	19.00 ± 4.05
Neutrophils (Ths/µL)	4.23 ± 0.86	4.91 ± 0.25	4.16 ± 0.79	3.48 ± 0.53	3.95 ± 2.38	n. m.
LDH (U/L)	187.60 ± 17.91	181.4 ± 15.34	287.9 ± 79.32	333.5 ± 85.04	186.9 ± 30.47	n. m.
Leucocytes (Ths/µL)	7.36 ± 0.82	7.36 ± 0.68	7.03 ± 1.20	6.34 ± 0.75	7.21 ± 2.13	n. m.
Lymphocytes (Ths/µL)	2.28 ± 0.46	1.48 ± 0.39	1.75 ± 0.52	1.75 ± 0.36	2.34 ± 0.60	n. m.
PT (%)	91.12 ± 12.22	89.25 ± 13.96	89.88 ± 16.26	88.57 ± 11.60	89.57 ± 12.97	n. m.

ALT: alanine transaminase; AP: alkaline phosphatase; AST: aspartate aminotransferase; CRP: C-reactive protein; GGT: gamma-glutamyltransferase; LDH: lactate dehydrogenase; n. m. = not measured; PT: prothrombin time; Ths: thousand.

**Table 4 pharmaceuticals-17-00278-t004:** Course of the study.

	Before		After IP Infusion				
Time	Screening	Baseline	0 h, 0.25 h, 0.5 h, 0.75 h, 1 h, 1.5 h, 2 h, 3 h	4 h, 6 h, 8 h, 10 h, 12 h, 18 h, 36 h	24 h, 48 h	72 h	Follow-Up
	-7 Days	0 h					14 ± 3 Days
Informed consent	x						
Inclusion/exclusion criteria	x						
Demographic data	x						
Body weight, BP, HR, ECG	x					x	x
Physical examination	x					x	x
Body temperature	x	x		x	x	x	x
Blood safety parameters	x				x	x	x
Blood ML concentration		x	x	x	x	x	x
Infusion of IP		x					
UE/SUE/SUSAR		x	x	x	x	x	x
Compliance		x	x	x	x	x	x

BP: blood pressure, HR: heart rate, ECG: electrocardiogram, ML: mistletoe lectin, IP: investigational product, UE: unexpected event, SUE: severe unexpected event, SUSAR: suspected unexpected serious adverse reaction.

## Data Availability

Data is available by the corresponding author on reasonable request.

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
