# Peer review of "Pharmakokinetics of Mistletoe Lectins after Intravenous Application of a Mistletoe Product in Healthy Subjects"

_pharmaceuticals, 2024, doi:10.3390/ph17030278_

Round 1
Reviewer 1 Report
Comments and Suggestions for Authors
I have gone through the entire research article. The writing part is very impressive. The authors have written a good introduction. Maintained very good flow in the entire research article. The study design was also planned very well. In my opinion, the submitted research article should go through minor revision. The comments are mentioned below, please address them accordingly.
1)Please improve the quality of Figure-1.
2)Authors should explain why liver enzymes and toxicity have been raised.
3) Have authors explored to lower dose concentrations instead of 2000mg? If not, what was the rationale behind using 2000mg dose?
4)Introduction looks bit small. Would authors mind increasing introduction content?
Author Response
We would like to thank you very much for careful and thorough reading of this manuscript and for the thoughtful comments and constructive suggestions, which help us to improve the quality of this article. All requested changes have been highlighted in yellow. We appreciate your input, your advice and the fast peer review. Please find our point-by-point response below.
1)Please improve the quality of Figure-1.
Response: We improved the quality of Figure 1.
2)Authors should explain why liver enzymes and toxicity have been raised.
Response: We added a few words to the discussion regarding your comment (see line 156-164). Unfortunately, little is known so far about possible mechanisms of liver injury after mistletoe application.
3) Have authors explored to lower dose concentrations instead of 2000mg? If not, what was the rationale behind using 2000mg dose?
Response: Yes, we explored lower doses in previous clinical trials (eg. Ref. 10). The dose of 2000 mg was the highest tolerable dose in a previous study (see line 209-215). In order to detect mistletoe lectins in the nanogram range, we opted for the highest possible dose.
4)Introduction looks bit small. Would authors mind increasing introduction content?
Response: The introduction has been expanded to include some basic information about mistletoe (see line 30-36, line 44-48 and line 68-70).
Reviewer 2 Report
Comments and Suggestions for Authors
Mistletoe are a large group of obligate parasitic herbs which have been credited with medicinal values. The clear composition of the product being reported is highly vague which is the main reason for threrapeutic failure in the herbal products. such studies need to be reported to mak people aware of the slient side effects which genrally go unnoticed but I would appreciate if the authors can further substantiate their findings with relevant data or pharamcologial efects of the other possible phytoconstituents of mistletoe in the product
Comments on the Quality of English Languagecan be revised
Author Response
We would like to thank you very much for careful and thorough reading of this manuscript and your important comment. More information on the composition of the mistletoe product has been added. The following constituents of Helixor® products have been identified so far: mistletoe lectins (as also described in the introduction to the manuscript, see line 53-60), oligo- and polysaccharides (predominantly of ester derivatives of D-galacturonane), flavonoids and phenylpropanoids (caffeic acid, ferulic acid and sinapic acid and their degradation products; see line 204-207). There is little data on the phytoconstituents in mistletoe products, but we have now also addressed this important point in the discussion of our work (see line 155-164).
Reviewer 3 Report
Comments and Suggestions for Authors
The study presented in this manuscript investigates the pharmacokinetics and safety profile of mistletoe lectins (ML) when administered intravenously in healthy subjects. The authors conducted a single-infusion trial with 2,000 mg of Helixor® P, a mistletoe product containing ML. The results indicate successful systemic absorption of ML, with a mean half-life of serum ML of 7.02 ± 2.01 hours. However, the study was terminated prematurely due to elevated liver enzymes in the participants. The authors concluded that the dose of 2,000 mg Helixor P® caused transient liver injury and should not be used for initial patient treatment.
Overall, the study provides valuable insights into the pharmacokinetics and safety of intravenously administered mistletoe lectins. However, there are a few areas that require further clarification and improvement before the manuscript can be considered for publication.
Major Comments:
-
The study included only eight healthy male volunteers. This small sample size raises concerns regarding the generalizability of the findings. It would be beneficial to provide a justification for the sample size selection and discuss the implications of this limitation on the study's outcomes. -
The premature termination of the study due to elevated liver enzymes is a significant finding. However, the authors did not provide a thorough discussion on the potential causes of liver injury and the implications for future clinical applications. It is crucial to explore the mechanisms underlying this adverse effect and discuss its clinical significance in more detail. -
The authors mentioned that the pharmacokinetics of natural mistletoe lectins are considerably longer than those of recombinant mistletoe. However, no direct comparison or reference to recombinant mistletoe lectins was included in the study. It is recommended to include a comparative analysis to strengthen this statement and provide a comprehensive understanding of the findings. - Minor Comments:
- Abstract and Introduction:
- The abstract and introduction sections should be revised to provide a more concise and clear overview of the study's objectives, methods, and key findings. The current abstract lacks specific details, such as the number of participants, and the introduction could benefit from a more comprehensive literature review on the use of mistletoe products in cancer treatment.
- Adverse Events:
- The authors briefly mentioned that potential side effects of Helixor® P infusion included headache, flu-like symptoms, and fever. It would be helpful to provide more specific information on the frequency and severity of these adverse events to better assess the safety profile of the mistletoe product.
- Recommendations for Future Research:
- Given the premature termination of the study and the observed transient liver injury, it would be valuable to include recommendations for future research. Suggestions for optimizing the dosing regimen, exploring potential mitigation strategies for liver injury, and investigating the efficacy of lower doses could enhance the practical implications of the study.
- In summary, the study presents important findings on the pharmacokinetics and safety of intravenously administered mistletoe lectins. However, several revisions and clarifications are needed to address the mentioned concerns and improve the manuscript's overall quality.
Author Response
We would like to thank you very much for careful and thorough reading of this manuscript and for the thoughtful comments and constructive suggestions, which help us to improve the quality of this article. All requested changes have been highlighted in yellow. We appreciate your input, your advice and the fast peer review. Please find our point-by-point response below.
Major Comments:
- The study included only eight healthy male volunteers. This small sample size raises concerns regarding the generalizability of the findings. It would be beneficial to provide a justification for the sample size selection and discuss the implications of this limitation on the study's outcomes.
Response: Thank you for this important concern. We had originally planned to include 12 subjects, based on the sample sizes of previous studies (see lines 268-271). Due to elevated liver enzymes we prematurely terminated the study after inclusion of 8 subjects (two groups of 4). The pharmacokinetics were very similar in all subjects, so that we conclude that the results are reliable despite the small number of cases. We have now addressed this issue as part of the discussion (line 137-140).
The premature termination of the study due to elevated liver enzymes is a significant finding. However, the authors did not provide a thorough discussion on the potential causes of liver injury and the implications for future clinical applications. It is crucial to explore the mechanisms underlying this adverse effect and discuss its clinical significance in more detail.
Response: Unfortunately, little is known so far about possible mechanisms of liver injury after mistletoe application. We added information to the discussion regarding your comment (see line 156-164).
The authors mentioned that the pharmacokinetics of natural mistletoe lectins are considerably longer than those of recombinant mistletoe. However, no direct comparison or reference to recombinant mistletoe lectins was included in the study. It is recommended to include a comparative analysis to strengthen this statement and provide a comprehensive understanding of the findings.
Response: The pharmacokinetics of intravenously applied recombinant mistletoe lectin was investigated by Schöffski et al., 2004. Although the design (dosing, cancer patients instead of healthy subjects) was not the same as in our study, we cited this reference (Ref. 21) for our statement, as pharmacokinetic parameters were obtained state of the art also in the Schöffski study. We have now formulated the conclusion more cautiously (“Pharmacokinetics of natural mistletoe appear to be considerably longer than that of recombinant mistletoe”, line 287).
Minor Comments:
- Abstract and Introduction:
The abstract and introduction sections should be revised to provide a more concise and clear overview of the study's objectives, methods, and key findings. The current abstract lacks specific details, such as the number of participants, and the introduction could benefit from a more comprehensive literature review on the use of mistletoe products in cancer treatment.
Response: We revised the abstract and the introduction (see line 30-36, line 44-48 and line 68-70).
- Adverse Events:
The authors briefly mentioned that potential side effects of Helixor® P infusion included headache, flu-like symptoms, and fever. It would be helpful to provide more specific information on the frequency and severity of these adverse events to better assess the safety profile of the mistletoe product.
Response: We have added some information on the side effects of mistletoe as part of the introduction (see line 44-48). Additionally, we have differentiated the flu-like symptoms in the results section: “Three subjects reported fatigue and flu like symptoms (malaise and “feeling like the beginning of a flu”) only in the first days after infusion of the IP which was possibly related to the study medication. None of the subjects developed fever. One subject suffered from headache.” (line 109-112)
- Recommendations for Future Research:
Given the premature termination of the study and the observed transient liver injury, it would be valuable to include recommendations for future research. Suggestions for optimizing the dosing regimen, exploring potential mitigation strategies for liver injury, and investigating the efficacy of lower doses could enhance the practical implications of the study.
Response: From our study we can only conclude that 2000 mg Helixor P intravenously should not be used for initial treatment (see line 168-174). Whether lower intravenous doses are safe and efficacious has to be investigated in future research. Furthermore, liver enzymes should be monitored when mistletoe preparations are given intravenously.
- In summary, the study presents important findings on the pharmacokinetics and safety of intravenously administered mistletoe lectins. However, several revisions and clarifications are needed to address the mentioned concerns and improve the manuscript's overall quality.
Response: Thank you again very much for your comments, which have really improved the manuscript.
Reviewer 4 Report
Comments and Suggestions for Authors
This study was addressed to evaluate pharmacokinetic indices and safety of mistletoe lectins (ML) in healthy male volunteers following intravenous infusion (within a 150 min-period) of 2,000 mg of the mistletoe products (MP) Helixor® P. Such pharmaceutical form is a watery extract [from fresh mistletoe (Viscum album L.) growing on pine trees] having the highest ML-content of all Helixor®-preparations. While being mean half-life of serum ML in the order of 7h, the study was terminated prematurely owing to a marked increase of liver enzymes (ALT, AST) although patients did not show significant clinical manifestations of liver disease and recovered completely. In conclusion, pharmacokinetics of natural mistletoe appeared to be longer than that of recombinant mistletoe, and the above dose of 2,000 mg of Helixor®P has to be avoided for initial patient treatment. In any case, liver enzymes must be kept under tight control when treating patients with intravenous Helixor®P.
Introduction underlines how MP may improve outcome of cancer patients and are approved for subcutaneous employment in various countries. However, there are not yet well defined pharmacokinetic indices regarding MP when given off label by intravenous (i.v.) route. In this regard, Authors report several studies in which Helixor® preparations (P, A and M ones) were used intravenously showing good tolerability according to various posological schemes (including i.v. doses ranging from 100 mg to 2,000 mg). In these studies, only minor side effects occurred with occasional appearance of also mild transient ALT elevation, pain and dyspnoea. ML (ML-I, -II and -III) represent major components of MP consisting of two chains connected by a disulfide bridge. A recombinant protein (rML) analogous to natural ML-I (nML) showed a short half-life of about 13 min in cancer patients. On the other hand, pharmacokinetics of nML have only been investigated following subcutaneous administration, being detectability of nML in serum considerably longer than that of rML.
Materials and Methods report with accuracy the adopted procedures [study design, recruitment, in- and exclusion criteria, use of the Helixor®P preparation, execution of the study (involving i.v. infusion of such preparation within a 150 min-period with next collection of blood samples until 72h for evaluating serum levels of ML], analytical method (ELISA) for determining these levels, clinical and haematochemical parameters, ethical approval, and the evaluated pharmacokinetic indices regarding Helixor®P [including t½(h), AUC etc.].
Results report the obtained evidences, i.e. biometric data of the eight included patients, concentration/time serum levels of ML, values of the considered pharmacokinetic, and haematochemical parameters. Four tables and one figure are included, being all explanatory and well prepared.
Discussion underlines how natural ML were shown to be provided with half-life longer (ca. 7h) than that of recombinant ML (13 minutes). Moreover, only at the high i.v. dose of 2,000 mg of natural ML (as present in commercial mistletoe preparations), there is occurrence of elevated serum levels of liver enzymes which, however, normalize enough rapidly without significant clinical outbreaks and/or other haematochemical alterations. In conclusion, liver enzymes, in particular, have to be carefully monitored in subjects given Helixor®P by i.v. route.
Overall, this study has been performed with diligent linearity and adequate methodology. Its interest consists in having shown that elevated i.v. doses of natural (mistletoe) ML are able to cause liver dysfunction with consistent elevation of serum ALT and AST. In any case, this elevation is transient and not accompanied by significant clinical outbreaks. Lexicon, sentence fluency, “English style”, tables, figure and references are adequate. Minimal editorial refinements might be desirable through the manuscript.
Comments on the Quality of English Language
The quality of English is adequate.
Author Response
We would like to thank you very much for careful and thorough reading of this manuscript and your positive evaluation!
Round 2
Reviewer 3 Report
Comments and Suggestions for Authors
Accepted